# A Retrospective Evaluation of the Predictive Value of Newborn Screening for Vitamin B12 Deficiency in Symptomatic Infants Below 1 Year of Age

**DOI:** 10.3390/ijns8040066

**Published:** 2022-12-14

**Authors:** Ulf Wike Ljungblad, Morten Lindberg, Erik A. Eklund, Ingjerd Sæves, Carlos Sagredo, Anne-Lise Bjørke-Monsen, Trine Tangeraas

**Affiliations:** 1Institute of Clinical Medicine, University of Oslo, P.O. Box 1171, Blindern, 0318 Oslo, Norway; 2Department of Pediatrics, Vestfold Hospital Trust, P.O. Box 1068, 3103 Tønsberg, Norway; 3Department of Medical Biochemistry, Vestfold Hospital Trust, P.O. Box 1068, 3103 Tønsberg, Norway; 4Department of Pediatrics, Clinical Sciences, Lund, Lund University, 221 84 Lund, Sweden; 5Norwegian National Unit for Newborn Screening, Division of Pediatric and Adolescent Medicine, Oslo University Hospital, 0424 Oslo, Norway; 6Department of Pharmacology, Division of Laboratory Medicine, Oslo University Hospital, 0424 Oslo, Norway; 7Laboratory of Medical Biochemistry, Innlandet Hospital Trust, 2609 Lillehammer, Norway; 8Laboratory of Medical Biochemistry, Førde Central Hospital, 6812 Førde, Norway; 9Department of Medical Biochemistry and Pharmacology, Haukeland University Hospital, 1400 Bergen, Norway

**Keywords:** vitamin B12 deficiency, homocysteine, infant, newborn screening, nitrous oxide, second-tier, vitamin B12

## Abstract

Background: The sensitivity of newborn screening (NBS) in detecting infants that later develop symptomatic vitamin B12 deficiency is unknown. We evaluated the predictive value using NBS algorithms in detecting infants that later were clinically diagnosed with symptomatic B12 deficiency. Furthermore, we investigated whether being born in a hospital using nitrous oxide (N_2_O) as pain relief in labor may have had an impact on total homocysteine at NBS. Methods: We retrospectively retrieved NBS data and analyzed total homocysteine, methylmalonic acid and methyl citrate on stored NBS dried blood spots (DBS) of 70 infants diagnosed with symptomatic B12 deficiency and compared them to 646 matched and 434 unmatched DBS controls to evaluate the Austrian and Heidelberg B12 NBS algorithms. Results: The sensitivity of NBS in detecting infants later diagnosed with symptomatic B12 deficiency at median age 10.9 weeks was ≤10%. Total homocysteine was higher in DBS for the unmatched controls who were born in hospitals providing N_2_O compared to in hospitals not providing N_2_O, with median total homocysteine 4.0 µmol/L compared to 3.5 µmol/L (n = 434, 95% CI 0.04–0.87, *p* = 0.03). Conclusion: NBS algorithms were unable to identify most infants diagnosed with symptomatic B12 deficiency after the neonatal period. Being born in hospitals providing N_2_O may impact total homocysteine at NBS.

## 1. Introduction

Vitamin B12 (B12) is important for neurodevelopment and even moderate deficiency during the first months of life may cause disease with tremor, apneas, seizures, and developmental delay [1,2]. Prompt B12 substitution effectively resolves the deficiency [3], but severe long-standing B12 deficiency may result in long-term neurological disabilities even if treated [4]. A higher incidence of B12 deficiency in newborn screening (NBS) programs have recently been demonstrated after the implementation of algorithms specifically designed for this purpose and when remethylation disorders have been introduced as primary targets of the NBS programs [5,6]. Total homocysteine (tHcy) is recognized as the best marker of B12 deficiency in this age group [2]. The B12 deficiency NBS algorithms published from Austria [6] and Heidelberg [5] utilized first and second-tier markers deriving from both B12-dependent pathways. Propionylcarnitine (C3) with different ratios and methylmalonic acid (MMA)/methylcitrate (MCA) were primary and secondary markers from the conversion of methylmalonyl-CoA to succinyl-CoA-pathway, whereas methionine with its ratio to phenylalanine and tHCy were first-tier and second-tier tests emanating from the remethylation of homocysteine to methionine [5,6,7,8,9]. These studies reported a positive predictive value of 67–81% [6] and 45% [5] using B12, holotranscobalamin (holoTC), tHcy and MMA to confirm the biochemical diagnosis of B12 deficiency. Since NBS for B12 deficiency mainly reveals maternal B12 deficiency, recognized as a main risk factor for infant B12 deficiency, it has the potential not only to detect the still asymptomatic newborn, but also the mother, allowing both to be treated and thus preventing symptoms and deficiency in the next pregnancy [5]. In Canada, 5% of women in fertile age has been found to have vitamin B12 deficiency [10]. Nitrous oxide (N_2_O), commonly used as pain relief in labor, accumulates in the fetus and is known to irreversibly inhibit methionine synthase by oxidizing the cobalt atom in a dose–response manner [11,12,13,14,15].

The aims of this study were to evaluate the Austrian and Heidelberg NBS algorithms applied retrospectively for infants clinically diagnosed with symptomatic vitamin B12 deficiency, and to assess if the availability of N_2_O, and thus its possible use as pain relief during labor at hospital of birth, could affect the NBS interpretation.

## 2. Materials and Methods

### 2.1. Study Population

We performed a case–control study with a group of symptomatic B12 deficiency cases and two groups of controls (Figure 1). We included infants below one year of age, born between 2011 and 2018, that were diagnosed and treated for symptomatic B12 deficiency. The treating physician decided upon B12 deficiency diagnosis from clinical symptoms and findings, and B12 status without any predefined criteria. These infants were designated as clinical cases and were identified after search for the International Classification of Disease 10 codes E53.8, E53.9, Z03.3, P90, P91.8, P28.4, R56, R58.8 or D51, with a concomitant B12 status analysis in medical record databases of two hospitals in the South-East of Norway [1]. We recruited a cohort of healthy, age-matched infants, referred to as clinical controls, scheduled for postnatal clinical follow-up in 2018–2019 from the Postnatal and Neonatal Unit at Vestfold Hospital Trust, Norway [16]. Details on inclusion, background characteristics, clinical and biochemical findings have been published elsewhere [1,15,16,17,18]. We also included NBS dried blood spot (DBS) controls, matched for date of birth, age in days, sex, hospital, birth weight and gestational age of the clinical cases and clinical controls, designated as matched NBS DBS controls. Additionally, another 450 unmatched NBS DBS controls were collected in 2020–2021 (Figure 1). The included hospitals were stratified according to the availability of nitrous oxide as pain relief during delivery.

### 2.2. Newborn Screening Analyses

Blood samples were collected on filter cards 48–72 h after birth and sent by prioritized mail to the Norwegian National NBS laboratory at Oslo University Hospital [16]. Only infants born after the expansion of the NBS program in Norway, on 1 March 2012, were included, as DBS before this date were destructed in accordance with Norwegian law. First-tier analyses of acylcarnitines were performed using the NeoBase 2 Non-Derivatized MSMS Kit (PerkinElmer, Turku, Finland) on an Acquity UPLC coupled to a Xevo TQS-micro mass spectrometer (Waters, Milford, MA, USA), after being punched (3.2 mm disc) with a Panthera-Puncher 9 (PerkinElmer, Turku, Finland). After the standard NBS analyses were performed, the DBS were first kept at +2–4 °C, for 1–3 months, followed by storage in a biobank at −20 °C until the second-tier analyses of tHcy, MMA and MCA were undertaken twice during 2020–2021. The second-tier analyses were performed at the time of the standard NBS analyses for the unmatched controls. A combined method for second-tier analysis of tHcy, MMA and MCA in DBS was set up using an LC-MS/MS method described elsewhere [15], and systematically introduced as second-tier analysis for cystathionine β-synthase deficiency and methylmalonic- and propionic aciduria in 2020. Readings without a tHcy peak were considered unreliable and therefore excluded. We used the previously published flowcharts from the Austrian NBS program [6] and the Heidelberg NBS program [5] to retrospectively categorize our study cohort’s NBS results into NBS positive or NBS negative B12 deficiency. We entered absolute NBS values from our own program corresponding to their suggested percentile-cutoffs [19]. We calculated the cutoff values for tHcy equivalent of the percentiles used by Rozmaric et al. [6] from the unmatched controls. We could not calculate the 99.9 percentile for MMA used by Gramer et al. [5] due to insufficient number of controls, and we therefore chose to use their absolute cutoff value. We compared matched controls to cases since DBS tHcy increased 0.35 µmol/L per year with storage time as shown previously. DBS MMA was not affected by storage [15].

### 2.3. Statistics

Continuous variables are presented as mean and standard deviation or if skewed, as median and interquartile range (IQR). Categorical variables are given as proportions and percentages and are compared between groups using the Fisher’s Exact test. Differences between independent groups are quantified with *t*-tests. We use receiver operating characteristic (ROC) curves with being a ‘symptomatic case’ as outcome variable to test the NBS analytes’ performance as classifiers. All statistical tests are two-sided, and a *p*-value < 0.05 is considered statistically significant. We present data for cases and controls where the combined results from expanded NBS and from second-tier analyses (Figure 1) are available. Data analyses were performed in IBM SPSS Statistics version 28 (IBM Inc., New York, NY, USA).

## 3. Results

During the study period 35,639 children were born in the catchment area. By the search string presented in the methods we identified 394 infants < 1 year. Of these, 130 were diagnosed and treated for B12 deficiency (130/35,639, 0.36%) and in 264 infants, B12 deficiency was not diagnosed. We invited 123 of the infants diagnosed with B12 deficiency [1], of which 93 infants were recruited and 30 did not reply or declined the invitation. We excluded 8 infants due to age over 1 year (n = 1), severe asphyxia (n = 1), genetic disease (n = 5) or no B12 deficiency (n = 1, erroneously included) [1]. Of the remaining 85 infants (Figure 1), nine were diagnosed presymptomatically with B12 deficiency, and six infants were excluded due to missing tHcy analyses, five because DBS had been destroyed and one case was born before the expanded NBS was introduced (Figure 1). Thus, 70 infants with symptomatic B12 deficiency were included for analyses in the present study.

At work-up, median [IQR] age was 10.9 [4.7–18] weeks, and the symptomatic B12 deficient cases had median [IQR] S-B12 197 [148–249] pmol/L, S-tHcy 12 [10–15] µmol/L, and S-MMA 1.50 [0.51–2.60] µmol/L. Twenty-eight (40%) had either S-B12 < 148 pmol/L or S-holoTC < 35 pmol/L, 34/67 (51%) had S-B12 < 200 pmol/L and 62/70 (89%) had either S-B12 < 200 pmol/L or S-tHcy > 10 µmol/L. Sixty of 66 (91%) had tHcy ≥ 8 µmol/L The mothers (n = 60) had median S-B12 254 pmol/L [187–342]. In a subgroup of 30 infants with S-B12 < 160 pmol/L (n = 20) or holoTC < 35 pmol/L (n = 10), median S-B12 was 144 [129–188] pmol/L, S-holoTC 31 [25–39] pmol/L, S-tHcy 12 [11–16] µmol/L, S-MMA 1.34 [0.43–2.42] µmol/L and the median of 25 maternal S-B12 was 229 [183–287] pmol/L.

We applied the same percentiles for tHcy as the Austrian published NBS algorithm [6]. The tHcy 89.2 percentile and the 96.7 percentile in the unmatched control group (n = 434) corresponded to 6.3 µmol/L and 8.6 µmol/L, respectively. The unmatched NBS controls were collected from 34 different hospitals with maternity wards, and N_2_O was available as birth analgesia at 25 (74%) of these hospitals. In total, 239/434 (55%) of unmatched controls were born in hospitals providing N_2_O. tHcy was higher for the unmatched controls who were born in hospitals providing N_2_O compared to in hospitals not providing N_2_O, with tHcy = 4.0 µmol/L compared to 3.5 µmol/L (n = 434, *p* = 0.03), while mean MMA was 0.26 µmol/L compared to 0.21 µmol/L, respectively (*p* = 0.131). The clinical cases and the matched controls were all born at two hospitals which provided N_2_O [15]. Descriptive characteristics are presented in Table 1 and Table 2. Clinical presentation and findings in cases and controls are presented in Table 3. None of the cases or controls were diagnosed with an inherited disorder of cobalamin metabolism.

First-tier pathways identified clinical cases in 19% using the Heidelberg algorithm [5] and 5.7% when incorporating the Austrian algorithm [6]. In a subgroup analysis restricting B12 deficiency to clinical cases with B12 < 160 pmol/L or holoTC < 35 pmol/L the ratio increased to 30% and 10%, respectively, using the above algorithms. For the matched controls, 14% were identified according to the Heidelberg algorithm and 4.5% using the Austrian algorithm, whereas for unmatched controls the corresponding proportions were 20% and 8.8%, respectively. When adding the second-tier analytes, the Heidelberg algorithm identified three clinical cases (4.3%), all three also identified in the subgroup (10%), compared to 0.6% and 0.7% of the matched and unmatched controls, respectively. When tHcy > 6.3 μmol/L was applied as second-tier cutoff-limit, the Austrian algorithm identified two clinical cases (2.9%), both to be found in the subgroup (6.7%), whereas 1.1% and 0.2% of matched and unmatched controls would have been subjected to repeat DBS, respectively. When tHcy > 8.6 μmol/L was attempted as second-tier cutoff, the Austrian algorithm did not identify any of the clinical cases but 0.2% of both matched and unmatched controls (Appendix A, Table 4 and Appendix A).

C3/C2 had the strongest correlation with plasma or serum tHcy at diagnosis of clinical B12 deficiency with r = 0.225 (*p* < 0.001, Table 5) and had the best diagnostic accuracy among the first-tier tests. C3/C2 correlated with NBS second-tier tHcy (r = 0.293, *p* < 0.001). Of the second-tier tests, tHcy had the strongest correlation (r = 0.492, *p* < 0.001) with serum or plasma tHcy at diagnosis of symptomatic B12 deficiency. NBS tHcy had the best diagnostic accuracy among the second-tier tests with AUC = 0.665, followed by MMA with AUC = 0.639. Methionine and methionine/phenylalanine did not correlate with diagnostic markers (Figure 2, Table 5 and Table 6).

## 4. Discussion

Our study showed that NBS markers failed to identify ≥90% infants diagnosed with symptomatic B12 deficiency after the newborn period. Restricting B12 deficiency to clinical cases with B12 < 160 pmol/L or holoTC < 35 pmol/l did not increase the sensitivity of NBS algorithms substantially. We also indirectly showed that N_2_O could interfere with the interpretation of second-tier NBS tHcy. It is generally agreed that tHcy is the best functional test for B12 deficiency in this age group, but the specificity is suboptimal as several of the published NBS algorithms contain a second DBS to show the persistence of elevated tHcy before the infant is recalled for confirmatory testing [5,6,20]. We propose N_2_O given as birth analgesia is one of the confounding factors that transiently increases tHcy. tHcy returns to the outset when the methionine synthase enzyme activity has been restituted by re-synthesis. This process requires B12, rendering mothers and fetuses with low B12 stores prone to B12 deficiency [1,15]. Our results confirmed the reservation made by Gramer et al. [5] that B12 deficiency presenting after the neonatal period is poorly detectable at NBS. Thus, our study adds to the discussion of the relevance and feasibility of including B12 deficiency as a primary target in NBS [21].

When authors of published NBS programs have reported high sensitivities and specificities for infant B12 deficiency [5,6], a biochemical definition of B12 deficiency on blood tests drawn at recall at median 4.5 weeks of age have been applied [6] and all cases have been reported to be symptom free [5,6]. Symptomatic infant B12 deficiency has been shown to manifest later than the first month of life [1,22,23,24] probably because most infants have sufficient B12 stores to remain asymptomatic the first month(s) of life. Further, there is a large discrepancy between the prevalence reported from NBS programs compared to the clinical settings: The birth prevalence of B12 deficiency reported from NBS programs are in the magnitude 0.01–0.09% [6,25]. In the southeastern part of Norway, a retrospective study found that 0.36% of infants under 1 year were diagnosed with B12 deficiency [1], while a Swedish retrospective study estimated an incidence of 0.31% [23]. Moreover, 10% of presumably healthy infants had mild symptoms and biochemical findings suggestive of B12 deficiency in a prospective study [16]. About two thirds of mainly breastfed infants below the age of six months have a biochemical profile indicative of vitamin B12 deficiency, which responds to B12 supplementation [26]. Intervention studies have shown that B12 supplementation to moderately B12 deficiency infants may improve both motor function and regurgitations, which suggests that an adequate B12 status is important for a rapidly developing nervous system [3]. There seems to be a ten times increase in infant B12 deficiency incidence depending on the diagnostic viewpoint: NBS, selective testing, or clinical screening. Theoretically then, our finding of a rather low, ≤10% sensitivity for NBS to identify symptomatic B12 deficiency was expected. Other risk factors beyond maternal B12 deficiency may come into play for infants with B12 deficiency during the first year of life. In the present study, we found associations between symptomatic B12 deficiency in infants with B12 < 160 or holoTC < 35 pmol/L and single parenthood, lack of employment, lack of B12 supplementation, known maternal B12 deficiency, self-reported hyperemesis, and exclusive breastfeeding. Of the five infants identified retrospectively with a positive NBS, four were exclusively breastfed, and the fifth infant had only recently been introduced to porridge after exclusive breastfeeding. NBS detects prenatal, maternal B12 deficiency. Breastfeeding is a postnatal risk factor. Infants to B12 deficient mothers are first born with diminished B12 stores and then fed with milk that contains less B12 [27]. Thus, the risk identified with NBS is propagated through exclusive breastfeeding. We have previously shown that formula feeding was protective of infant B12 deficiency [1,15,16], so if the infant is formula fed, this chain of risks is broken, and the predictability of NBS for infant B12 deficiency is lost. This is unique for B12 deficiency NBS. In no other disease screened for is the source feeding the only factor decisive for symptom presentation. Another factor may be maternal use of N_2_O during labor, a common form of pain relief. We found that N_2_O was provided as an analgesia option at 74% of the hospitals from where the un-matched controls were collected, in a distribution representative for Norway. We have previously shown that N_2_O was used by 64–68% of women in labor [1]. In the present study, we showed that tHcy was higher in newborns at hospitals where N_2_O was optionable as birth analgesia compared to where N_2_O was unavailable. Previously, we found the maternal dose of N_2_O to be a significant predictor for tHcy (but not MMA) at NBS. However, at diagnosis of symptomatic infant B12 deficiency, maternal dose of N_2_O was associated with both tHcy and MMA. We therefore suggested that N_2_O is a risk factor for later presenting symptomatic infant B12 deficiency [1,15,16].

Presenting symptoms in three of the five NBS positive infants were spells of apneas, absences, or seizures, and two of five showed abnormal eye contact. These are potentially life-threatening symptoms that could have been prevented with NBS for B12 deficiency. Half of the cases with B12 < 160 or holoTC < 35 pmol/L had head lag at pull-to-sit and a third had tremor, which were significant findings compared to clinical controls. The yield of NBS was doubled in this subgroup with a stricter B12 definition, although the sensitivity remained ≤10%. We speculate in a difference in sensitivity of having symptoms from low B12 between different genotypes of the B12 dependent enzymes, and then there is a risk that the more sensitive and vulnerable infants will be missed in NBS.

Our study was original in the design of combining clinical cases with symptomatic B12 deficiency with their respective NBS results and re-analysis of DBS. In the unmatched 450 DBS controls, we only had access to whether N_2_O was available at the hospital of birth or not; however, data on the individual mothers receiving N_2_O or not was not retrieved for this cohort. This information would probably have strengthened the association between mothers N_2_O intake and tHcy in DBS as we have shown in a recent study [15]. Maternal B12 parameters were neither available during pregnancy nor at birth, representing a limitation to our study. Additionally, as we previously showed, tHcy increases with storage time of DBS [15], and this introduced a bias to our cohort. This may, theoretically, have overestimated some of the few oldest cases picked up by the Austrian algorithm second-tier tHcy test [6], but it would not change the conclusion of our study.

## 5. Conclusions

To summarize, NBS showed a low sensitivity for symptomatic B12 deficiency in our cohort of infants presenting beyond the neonatal period. However, NBS may still play an important role in detecting and treating breastfed newborns with B12 deficiency but the shortcomings of NBS in detecting all infants prone to develop B12 deficiency should be acknowledged and sustain awareness of B12 deficiency as a cause of subtle and overt neurological symptoms in infancy.

## Figures and Tables

**Figure 1 IJNS-08-00066-f001:**
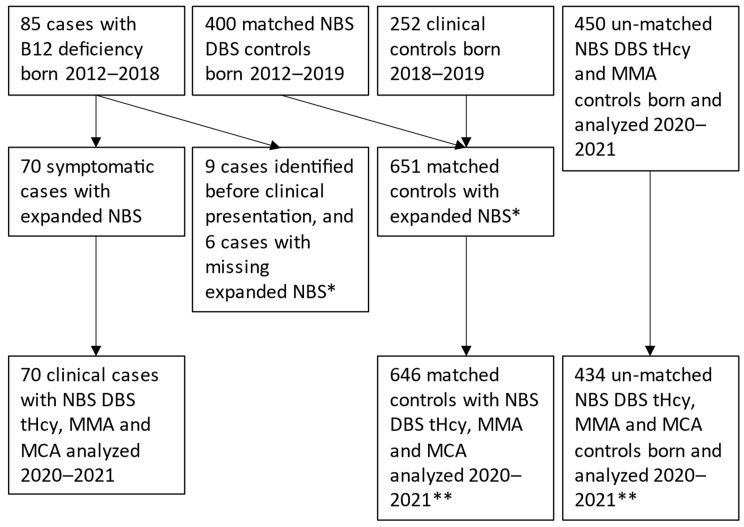
Inclusion of clinical cases and controls, * = missing infants born before 2012, missing expanded NBS, ** = missing infants with unsuccessful 2nd tier analyses, NBS = newborn screening, DBS = dried blood spot, tHcy = total homocysteine, MMA = methylmalonic acid, MCA = methyl citric acid.

**Figure 2 IJNS-08-00066-f002:**
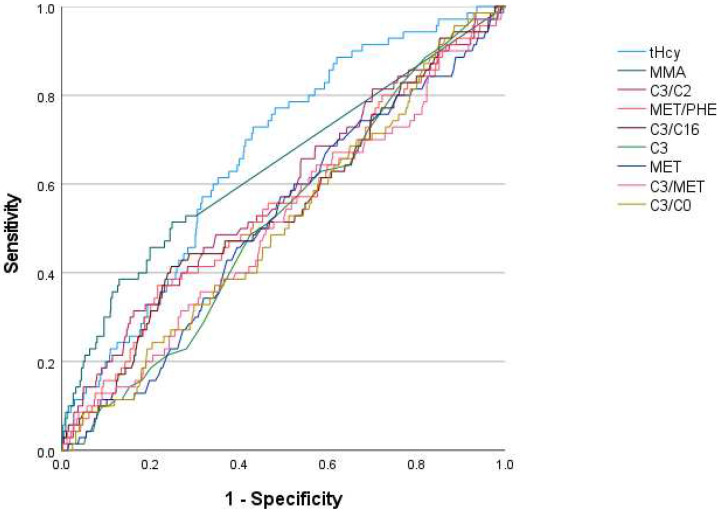
ROC curves of newborn screening parameters’ diagnostic accuracy of being a case with B12 deficiency, cases and matched controls, n = 716. MET = methionine, PHE = phenylalanine, C0 = carnitine, C2 = acetylcarnitine, C3 = propionylcarnitine, C16 = palmitoylcarnitine, tHcy = total homocysteine, MMA = methylmalonic acid.

**Table 1 IJNS-08-00066-t001:** Descriptive characteristics of cases and controls, mean (SD) and n (%).

	Positive NBS (n = 5)	Clinical Cases with B12 < 160 or holoTC < 35 pmol/L (n = 30)	Clinical Cases (n = 70)	Clinical Controls (n = 252)	Matched Controls (n = 646)	Unmatched Controls (n = 434)
Gestational age (weeks)	38 (4)	39 (2)	39 (2)	39 (2)	39 (2)	39 (2)
Birthweight (grams)	3152 (840)	3327 (554)	3401 (627)	3296 (666)	3427 (588)	3493 (540)
NBS DBS age (hours)	59 (15)	59 (10)	59 (14)	62 (16)	62 (15)	57 (18)
DBS storage time (years)	3.7 (1.5)	4.0 (1.8)	4.0 (1.8)	1.9 (0.2)	2.6 (1.6)	0
Female	2 (40)	12 (40)	29 (41)	125 (50)	301 (47%)	216 (50%)

NBS = newborn screening, DBS = dried blood spot.

**Table 2 IJNS-08-00066-t002:** Descriptive characteristics of clinical cases and controls, mean (SD) and n (%).

	Positive NBS (n = 5)	Clinical Cases with B12 < 160 or holoTC < 35 pmol/L (n = 30)	Clinical Cases (n = 70)	Clinical Controls (n = 252)	Difference Compared to Clinical Controls (Fisher’s Exact Test or *t*-Test, *p*)	
					Positive NBS	Clinical Cases with B12 < 160 or holoTC < 35 pmol/L
Married/cohabitant	5 (100)	27 (90)	63 (90)	249 (99)	1.0	**0.01**
Higher education	4 (80)	20 (67)	45 (64)	169 (69)	1.0	0.84
Origin outside the Nordic countries	1 (20)	7 (23)	11 (16)	53 (21)	1.0	0.81
Employment last 2 years	4 (80)	20 (77)	50 (76)	220 (91)	0.40	**0.045**
Smoking last 2 years	0	5 (17)	9 (13)	30 (12)	1.0	0.56
Meat-eater	5 (100)	29 (97)	69 (99)	241 (97)	0.13	0.60
Known maternal B12 deficiency	1 (20)	7 (24)	17 (25)	24 (9.7)	0.41	**0.03**
Celiac disease	0	1 (3.3)	5 (7.1)	8 (3.2)	1.0	1.0
Primipara	4 (80)	15 (50)	30 (43	138 (55)	0.38	0.70
Diabetes in pregnancy	0	1 (3.3)	3 (4.3)	16 (6.3)	1.0	1.0
Metformin use	0	1 (3.6)	2 (3.4)	9 (3.6)	1.0	1.0
Hyperemesis (self-reported)	3 (60)	14 (47)	23 (33)	67 (27)	0.13	**0.03**
Folate during pregnancy	5 (100)	25 (83)	56 (81)	219 (88)	1.0	0.56
B12 containing supplement during pregnancy	3 (60)	11 (37)	28 (41)	163 (65)	1.0	**0.005**
Preeclampsia	0	2 (6.7)	4 (5.8)	14 (5.6)	1.0	0.68
N_2_O analgesia	4 (80)	20 (67)	43 (62)	170 (68)	1.0	1.0
Cesarian section	0	5 (17)	13 (19)	56 (22)	0.59	0.64
Female	2 (40)	12 (40)	29 (41)	124 (49)	1.0	0.44
Multiple birth	0	2 (6.7)	2 (2.9)	29 (12)	1.0	0.55
Preterm GA 32–36 weeks	1 (20)	4 (13)	6 (8.6)	43 (17)	1.0	0.80
Small for GA < 10p	1 (20)	3 (10)	10 (14)	46 (18)	1.0	0.32
Exclusively breastmilk	4 * (80)	23 (79)	49 (72)	82 (33)	**0.047**	**<0.001**
Yearly household income (NOK)	742,800 (375,312)	860,960 (392,077)	894,293 (329,007)	971,884 (341,984)	0.14	0.14
Mother’s BMI before pregnancy	22.8 (3.7)	25.1 (6.5)	24.8 (5.5)	24.7 (5.0)	0.39	0.74
Mother’s age at birth	26 (3.9)	31 (4.1)	31 (4.3)	30 (4.7)	0.06	0.43
Dose N_2_O ** (min × conc)	85 (83)	71 (105)	63 (90)	62 (81)	0.54	0.58
Gestational age in weeks	38.3 (3.7)	39.1 (2.4)	39.3 (2.5)	39.1 (2.2)	0.46	0.99
Birthweight z-score	−0.32 (1.31)	−0.40 (1.06)	−0.28 (1.12)	−0.41 (1.20)	0.86	0.96
Infant age in weeks	14.3 (8.0)	16.7 (11.8)	13.5 (10.7)	20.8 (5.2)	0.007	0.001
Weight z-score	−0.34 (1.25)	−0.51 (1.23)	−0.46 (1.15)	−0.09 (1.06)	0.6	0.06

* The single case not exclusively breastfed had recently introduced porridge in addition to breastmilk. ** Dose of nitrous oxide (N_2_O) is the product of concentration of N_2_O and the intermittent administration time in minutes. NBS = newborn screening, NOK = Norwegian krone, BMI = Body Mass Index. Significant *p*-values (<0.05) are written in bold.

**Table 3 IJNS-08-00066-t003:** Clinical symptoms and findings of cases and controls, n (%).

	Positive NBS (n = 5)	Clinical Cases with B12 < 160 or holoTC < 35 pmol/L (n = 30)	Clinical Cases (n = 70)	Clinical Controls (n = 252)	Difference Compared to Clinical Controls (Fisher’s Exact Test, *p*)	
					Positive NBS	Clinical Cases with B12 < 160 or holoTC < 35 pmol/L
Spells (motor seizures, apneas, or absences)	3/5 (60)	10/23 (43)	29/60 (48)	0/250 (0)	**<0.001**	**<0.001**
Tremor	1/4 (25)	8/22 (36)	20/58 (34)	13/250 (5.2)	0.20	**<0.001**
Irritability	1/4 (25)	4/21 (19)	10/56 (18)	19/252 (7.5)	0.28	0.09
Head lag at pull-to-sit	2/4 (50)	9/18 (50)	23/44 (52)	38/250 (15)	0.12	**0.001**
Abnormal eye contact	2/5 (40)	4/22 (18)	7/54 (13)	0/250 (0)	**<0.001**	**<0.001**

NBS = newborn screening. Significant *p*-values (<0.05) are written in bold.

**Table 4 IJNS-08-00066-t004:** Results from applying B12 deficiency algorithms according to the Austrian and Heidelberg NBS for cases and controls (n, %).

	Clinical Cases with B12 < 160 pmol/L or holoTC < 35 pmol/L (n = 30)	Clinical Cases (n = 70)	Matched Controls (n = 646)	Un-Matched Controls (n = 434)
Heidelberg 1st tier positive	9 * (30%)	13 * (19%)	93 (14%)	85 (20%)
Heidelberg 1st and 2nd tier positive	3 ** (10%)	3 ** (4.3%)	4 (0.6%)	3 (0.7%)
Austrian 1st tier positive	3 * (10%)	4 * (5.7%)	29 (4.5%)	38 (8.8%)
Austrian 1st tier positive and tHcy > 6.3	2 *** (6.7%)	2 *** (2.9%)	7 (1.1%)	1 (0.2%)
Austrian 1st tier positive and tHcy > 8.6	0	0	1 (0.2%)	1 (0.2%)

* = one in C3 pathway, the rest in MET pathway ** = only MET pathway, *** = one in C3 and MET pathways, respectively.

**Table 5 IJNS-08-00066-t005:** Univariate correlations between tHcy, MMA, B12 at mean (SD) 19 (7.4) weeks of age and NBS parameters at mean (SD) age 62 (15) hours of age (Pearson, n = 316–317).

Newborn Screening Parameter	tHcy µmol/L	MMA µmol/L	S-Vitamin B12 pmol/L
tHcy	0.492 **	0.275 **	−0.208 **
MMA	0.235 **	0.187 **	−0.100
MET	−0.076	0.066	0.022
MET/PHE	−0.029	0.049	−0.044
C3	0.134 *	0.085	−0.117 *
C3/C2	0.225 **	0.165 **	−0.174 **
C3/C16	0.120 *	0.126 **	−0.069
C3/MET	0.178 **	0.065	−0.143 *
C3/C0	0.094	0.001	−0.106

* = correlation significant <0.05, ** <0.001, MET = methionine, PHE = phenylalanine, C0 = carnitine, C2 = acetylcarnitine, C3 = propionylcarnitine, C16 = palmitoylcarnitine, tHcy = total homocysteine, MMA = methylmalonic acid.

**Table 6 IJNS-08-00066-t006:** ROC areas under the curve, diagnostic accuracy of newborn screening parameters for being a case with B12 deficiency; clinical cases (n = 70), clinical cases with B12 < 160 or holotranscobalamin < 35 pmol/L (n = 30) and matched controls (n = 646).

Newborn Screening Variable	Area under the Curve	
	Clinical Cases	Clinical Cases with B12 < 160 pmol/L or holoTC < 35 pmol/L
tHcy	0.665	0.708
MMA	0.639	0.636
C3/C2	0.579	0.600
MET/PHE	0.550	0.579
C3/C16	0.547	0.525
C3	0.517	0.560
MET	0.515	0.542
C3/MET	0.512	0.544
C3/C0	0.510	0.515

MET = methionine, PHE = phenylalanine, C0 = carnitine, C2 = acetylcarnitine, C3 = propionylcarnitine, C16 = palmitoylcarnitine, tHcy = total homocysteine, MMA = methylmalonic acid.

## Data Availability

Not applicable.

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
