# Peer review of "A Retrospective Evaluation of the Predictive Value of Newborn Screening for Vitamin B12 Deficiency in Symptomatic Infants Below 1 Year of Age"

_2409-515X, 2022, doi:10.3390/ijns8040066_

Round 1

Reviewer 1 Report

This paper is a retrospective study of vitamin B12 deficiency (B12D), which is asymptomatic in the neonatal period and diagnosed in infancy after NBS, to determine whether it can be diagnosed from NBS laboratory data. In addition, the impact of nitrous oxide, which is used to reduce pain during labor and delivery, on the diagnosis of B12D has been studied in several cases and control cases. Although the effects of nitrous oxide on vitamin B12 metabolism are well known, I thought the report was original in its comparison of large number of clinical cases with the subject cases as one of the confounding factors in discussing the predictive potential of NBS for infant B12 deficiency. The statistical considerations of the authors in this paper are well organized and easy for the reader to understand, making this an excellent paper. I hope that my comment is useful for the improvement of the article.

1. Like References 5 and 6, this paper discusses the subject of B12D due to maternal vitamin B12 deficiency. In contrast to these, many type of cobalamin metabolism deficiencies (such as CblC deficiency) exist as B12D in NBS. It is presumed that the 70 clinical cases included in the study do not include cobalamin metabolism deficiencies that were not diagnosed without abnormal values at the time of NBS. If so, is there no need to clarify this point?

2. Footnote in Table 3:

Error: total homo-cysteine ⇒ Correction: total homocysteine

Author Response

Thank you so much for your comments and suggestions.

  1. Like References 5 and 6, this paper discusses the subject of B12D due to maternal vitamin B12 deficiency. In contrast to these, many type of cobalamin metabolism deficiencies (such as CblC deficiency) exist as B12D in NBS. It is presumed that the 70 clinical cases included in the study do not include cobalamin metabolism deficiencies that were not diagnosed without abnormal values at the time of NBS. If so, is there no need to clarify this point?

Response: A good point, thank you. We have added a sentence where this is clarified in the results: “None of the cases or controls were diagnosed with an inherited disorder of cobalamin metabolism”, line 165.

  1. Footnote in Table 3:

Error: total homo-cysteine ⇒ Correction: total homocysteine

Response: Corrected.

Reviewer 2 Report

The authors present compelling evidence on the lack of predictive value of newborn screening for detecting vitamin B12 deficiency in symptomatic infants below 1 year. The article is well written.

I have one question related to the discussion the authors state that feeding choice will only effect B12 deficiency however it is relevant to maternal carnitine deficiency as well and carnitine transporter deficiency is included in the list of diseases screened for in Norway

Discussion line 252: This is unique for B12 deficiency screening since no other 257 diseases screened for can be modified with the choice of feeding

Author Response

Response: Thank you, this is an interesting viewpoint, which renders the cited sentence of ours imprecise. However, whereas breastfed infants born to mothers with B12 deficiency may become symptomatic unless substituted, there is no compelling evidence in the literature that this is the case in a breastfed infant born to a mother with carnitine transporter deficiency (despite the serum free carnitine levels are lower if exclusively breastfed compared to formula fed).

We suggest modifying the sentence on lines 259-260 to: “This is unique for B12 deficiency screening. In no other disease screened for is the source of feeding paramount for symptom presentation”.